# Over-The-Counter Codeine: Can Community Pharmacy Staff Nudge Customers into Its Safe and Appropriate Use?

**DOI:** 10.3390/pharmacy8040185

**Published:** 2020-10-08

**Authors:** Sapana Mody, Charlotte L. Kirkdale, Tracey Thornley, Aimi Dickinson, Anthony J. Avery, Roger Knaggs, Sarah Rann, Ruth Bastable

**Affiliations:** 1Boots UK, Thane Road, Nottingham NG90 1BS, UK; charlotte.kirkdale@boots.co.uk (C.L.K.); tracey.thornley@boots.co.uk (T.T.); aimi.x.dickinson@boots.co.uk (A.D.); 2School of Pharmacy, University of Nottingham, University Park, Nottingham NG7 2RD, UK; roger.knaggs@nottingham.ac.uk; 3Division of Primary Care, School of Medicine, University of Nottingham, University Park, Nottingham NG7 2RD, UK; Anthony.avery@nottingham.ac.uk; 4Formerly CDAO East of England, NHS, Medical Directorate, Victoria House, Capital Park, Cambridge CB21 5XE, UK; sfrann@doctors.org.uk; 5HMP Littlehey, Huntingdon PE28 0SR, UK; rbbastable@aol.com; 6National Health Service, London SE1 6JW, UK; 7Royal College of General Practitioners, London NW1 2FB, UK

**Keywords:** behavioural nudge, codeine, community pharmacy, opioid misuse, over-the-counter

## Abstract

The misuse of opioids, including codeine which is sold over-the-counter (OTC) in United Kingdom (UK) community pharmacies, is a growing public health concern. An educational Patient Safety Card was developed and piloted to see if it nudged customers into the safe and appropriate use of OTC codeine. Exploratory analysis was conducted by (i) recording quantitative interactions for people requesting OTC codeine in community pharmacies; and (ii) a web-based pharmacy staff survey. Twenty-four pharmacies submitted data on 3993 interactions using the Patient Safety Card. Staff found the majority of interactions (91.3%) to be very or quite easy. Following an interaction using the card, customers known to pharmacy staff as frequent purchasers of OTC codeine were more likely not to purchase a pain relief medicine compared to customers not known to staff (5.5% of known customers did not purchase any pain relief product versus 1.1% for unknown customers (χ^2^ = 41.73, df = 1, *p* < 0.001)). These results support both the use of a visual educational intervention to encourage appropriate use of OTC codeine in community pharmacy and the principles behind better self-care.

## 1. Introduction

Opioid analgesics are used for pain relief where medicines such as paracetamol or ibuprofen alone have not provided relief [1]. In addition to being available on prescription in the UK, codeine-containing medicines are available to purchase from a pharmacy (usually behind the counter or in locked cabinets), sold by or under the supervision of a pharmacist, and are known by the metonymic term of over-the-counter (OTC). The total sales of all codeine-containing OTC products in the United Kingdom were 26 million units, or 7.8% of the total adult oral pain relief market [2].

Derived from the opium poppy, side effects of codeine consumption include drowsiness, respiratory depression and also psychoactive effects, such as feelings of euphoria and intoxication [3]. Codeine is available in co-formulations, including with paracetamol and ibuprofen, which have been associated with serious clinical complications such as life-threatening gastrointestinal bleeding and critical hypokalemia [4]. For children, while codeine is licensed for use in those aged over 12 years in the UK, it is contraindicated in those under 18 years who have breathing problems [5]. Regular or excessive use of codeine, even for short periods, can lead to tolerance [6], and psychological and physical dependence [5]. Those that become dependent may experience negative personal, professional or social impacts, and obtain illicit supplies; this could lead to criminality and has been linked into several preventable deaths [7,8,9]. Although OTC medicines abuse is an internationally recognized problem [10], the extent of codeine misuse is unclear due to the limited availability of data. One report on pharmacist perceptions indicated a perceived increase in misuse [11], and another cross-sectional survey of the general population reported that the products most commonly misused were codeine-containing analgesics [12].

The ubiquitous nature of OTC medicines enables consumers to take responsibility for their own health, offers convenience and choice, and reduces the burden on the health service (National Health Service, NHS) with its associated cost to taxpayers [13,14]. Availability of OTC medicines may result in a public perception of product safety, although there is conflicting evidence about individuals’ knowledge of possible harm and risks specifically in relation to medicines that could be misused [15,16,17]. Some individuals may unintentionally misuse codeine, for example, by taking it for longer than recommended (maximum of three days, maximum daily dose 240 mg) [18]. To compound the issue, there are multiple sources of information which individuals can consult and bypass healthcare professionals, for example on the internet [7,19]. In addition, patient information leaflets supplied with medicines can be complex and are often not consulted by consumers [15]. Leaflets also rely on the individual’s health literacy for comprehension, although poor health literacy is widespread and directly contributes to health inequalities [20,21].

In 2009, the Medicines and Healthcare products Regulatory Agency implemented new measures to minimize the risk of harm from OTC codeine (including reducing OTC codeine’s licensed indications, improving labelling (‘can cause addiction’ and ‘for three days use only’ to be prominently displayed on the pack) and limiting the pack size to 32 tablets) [22]. There is no legal limit on the sale of codeine-containing medicines, either in one transaction or over a set time period; however, the Royal Pharmaceutical Society recommends that only one pack of codeine- or dihydrocodeine-containing medication should be sold [23]. In 2018, Australia joined 25 other countries including Germany, Japan and the US in restricting codeine to prescription-only due to the health risks it posed [24,25,26]. This change has received mixed responses from pharmacists regarding their ability to support consumers with pain management, although it appears to have helped reduce codeine misuse [27,28].

In the UK, despite evidence supporting its use for short-term and self-limiting conditions [5], the limited evidence supporting analgesic benefit from codeine [29] means that misuse concerns remain. Reclassification of OTC codeine could become burdensome for the NHS, goes against NHS policy of promoting self-care [30], is likely to be met with resistance from manufacturers [14], and may result in codeine being sourced illicitly [31]. Some experts argue that an appropriate and enhanced intervention, at an early stage in community pharmacy, is preferable [26].

Pharmacists have a professional duty to support individuals’ health decisions [32] and could be more proactive when facing potential misuse [18,33,34]. Pharmacists are well placed to support consumers with responsible decision-making regarding medication [35], despite frequently cited barriers of resource and capacity constraints [36]. Provision of advice has been shown to support the selection of the most safe and effective product [37], improve health outcomes, and positively impact quality of life [38]. There is, however, evidence that appropriate advice is not always provided and can be of variable quality [39,40,41].

For OTC codeine, behavioral change techniques could be particularly effective for consumers more focused on the current benefit of analgesia, without consideration of the potential consequences from misuse [42]. According to Thaler and Sunstain, consumers who consider the ‘benefits now’ and ‘costs later’ can be ‘nudged’ into altering their behavior [43]. Incorporation of an educational nudge could be one method to improve responsible patient usage of OTC codeine. Therefore, a ‘Patient Safety Card’ was developed based on the framework that capability, opportunity and motivation can impact on behavior (‘COM-B’) [44]. The card was designed for community pharmacy staff to use with purchasers of OTC codeine medicines to reinforce and encourage its safe and appropriate use.

The aim of this evaluation was to pilot the card in pharmacies to see if it supported pharmacy teams’ interactions with consumers to influence (i.e., nudge) behavior towards the safe and appropriate use of OTC codeine. The objectives of this work were to: (1) test the ease with which pharmacy teams felt able to encourage the safe and appropriate use of OTC codeine by using the card alongside the normal provision of advice (in both known and unknown customers); (2) understand the views of pharmacy teams on customers’ receptiveness to the card; (3) assess if provision of the card led to an alternative outcome to the customer’s intended OTC codeine purchase; (4) explore the views and experiences of pharmacy teams on use of the card and any improvements that could be made to its format and design; and (5) solicit views on potential risk minimization measures relating to OTC codeine.

## 2. Materials and Methods

Using the principals of the Antimicrobial Resistance checklist [45] as a basis for the intervention, and the five clinical points considered key to the safe and appropriate use of OTC codeine [5], a panel of experts (including regulatory experts and community pharmacists with experience selling these medicines OTC) discussed and developed the final messages. As a result, an A6-sized card (Appendix A) was developed and distributed to 36 community pharmacies across a national chain (Boots UK). Pharmacies were purposefully selected based on the highest volume of OTC codeine sales, including a range of pharmacy formats (high street, local community and shopping centers), geographical spread and taking into account Covid-19′s operational impact.

The card was handed over to the patient by the pharmacy team member supporting the interaction (pharmacy advisor, pharmacy technician or pharmacist). It was used in addition to the pharmacy chain’s healthcare consultation model, which focusses on ensuring safe and appropriate sales, and empowering patient choice through a holistic, advice-led approach. After a suitability assessment, the consultation model supports provision of advice using the CARE framework (Counsel; Avoid; Read; Escalate). This framework includes counselling on how to take medication, any reason the medication should be avoided, a reminder to read the patient information leaflet, and advice for escalation actions if symptoms remain unresolved (safety netting). Codeine is also highlighted on a list of ‘people and products’ to take extra care with during an interaction, which complements the healthcare consultation model.

Pharmacy staff were asked to complete a paper copy of a data entry ‘tracker’ after each OTC codeine product request where the card was used over a five week period in June and July 2020 (Appendix A). Pharmacies were told to include transactions for codeine-containing medicines, including codeine and dihydrocodeine. The main fields recorded were (i) staff ease of use of the card; (ii) receptiveness of customer to the card; (iii) outcome of customer consultation; and (iv) an annotation where the customer was familiar to staff as a frequent purchaser of OTC codeine in that pharmacy (“known customer”). Data were collected (via electronic photos of the paper copies being sent to Boots head office staff), inputted and descriptive analysis performed using Microsoft Excel 365 version 16.37, and a chi squared test for association. Associations between receptiveness and ease of use and whether there were any differences in outcomes between known and unknown customers were explored.

A pharmacy staff survey was used to collect insights into experience of card use to see if it supported perceived behavioral change with customers, and solicit views on the wider management of OTC codeine (Appendix A). The survey was piloted on a small number of colleagues before finalizing the questions for use with participants. All pharmacy staff involved in using the card were invited to complete the pseudonymized online survey over a one week period after the pilot had begun. Invitations to participate were sent via the internal communications channels to all participating pharmacies. Data were manually coded to explore themes based on the questions posed using thematic analysis, and the COM-B framework [44]. The COM-B model proposes that a combination of the following impacts on behavior (B): C: capability (the individual’s physical and psychological ability, including their skill and knowledge); O: opportunity (external social or physical contextual factors); and M: motivation (an individual’s automatic and reflective processes, for example, their habits, choices and beliefs). By analyzing the three determinants of behavior, possible intervention functions based on Michie’s Behaviour Change Wheel can then be linked to specific behavioral change techniques (i.e., a ‘nudge’) [44]. An educational intervention suits circumstances where capability and motivation can be changed, and has been shown to be highly successful [46,47].

Approval for the service evaluation was obtained from Nottingham Business School, Nottingham Trent University (as analysis was completed as part of an MBA study), and Boots’ Research Governance Board.

## 3. Results

Of the 36 pharmacies involved in the project, 3993 complete tracker entries were received from 24 pharmacies. For these 24 pharmacies, there was a mean of 166 responses per pharmacy; range 8 to 480 overall responses per pharmacy; mean 10.9 responses per pharmacy per day, range 0 to 48 responses per pharmacy per day). Survey responses were received from 23 pharmacy staff across 22 pharmacies (10 pharmacists; 3 pharmacist store managers; 2 store managers; 4 manager assistants; 2 pre-registration pharmacists; 1 pharmacy advisor; and 1 pharmacy technician). Datasets are available in the Appendix A.

### 3.1. Tracker Data

Pharmacy staff considered the vast majority of conversations (91.3%, *n* = 3644) to be very or quite easy, and, of these, the majority of customers (83.4%, *n* = 3040) were reported to be receptive. Where pharmacy staff found the conversation difficult (*n* = 349), 90.0% of customers were not receptive or refused the card.

Known customers were observed to account for 8.7% (*n* = 347) of tracked interactions (Table 1). Pharmacy staff found 90.2% (*n* = 313) of conversations with known customers and 91.4% (*n* = 3331) of unknown customers to be easy. In terms of customer receptiveness, 72.9% (*n* = 253) of known customers and 77.4% (*n* = 2822) of unknown customers were perceived by the pharmacy staff to be receptive (χ^2^ = 3.61, df = 1, *p* = 0.06). Known customers refused the card in 11.0% (*n* = 38) of cases, compared to 6.6% (*n* = 239) of unknown customers (χ^2^ = 9.48, df = 1, *p* = 0.002).

The vast majority of customers (97.4%, *n* = 3891) purchased the originally intended OTC codeine product (Table 2); 94.5% (*n* = 328) for known customers and 97.7% (*n* = 3563) for unknown customers (χ^2^ = 13.03, df = 1, *p* < 0.001). Furthermore, 5.5% (*n* = 19) of known customers did not purchase any pain relief product, which was different to unknown customers at 1.1% (*n* = 40) (χ^2^ = 41.73, df = 1, *p* < 0.001).

### 3.2. Staff Survey

All pharmacy staff (pharmacists and healthcare staff) that responded to the survey reported using the card to support their conversations with customers when purchasing OTC codeine products. An overarching theme was that the card helped to reiterate the key risks associated with the use of OTC codeine with 91.3% (*n* = 21) of survey respondents saying it supported their conversations. All respondents replied yes (47.8%, *n* = 11) or maybe (52.2%, *n* = 12) when asked if they would recommend using the card to other pharmacy teams and a number commented that the messaging conveyed the key risks in a clearer and more readable way than the packaging or patient information leaflet. Another key theme was that customers were able to take the card away to read, review, refer to and reflect on at a later stage. Only a third of respondents (34.8%, *n* = 8) considered that the card would help achieve the most appropriate outcome for the customer. This appeared to be based on the fact that customers often requested codeine by name and, as noted previously, the vast majority of customers (97.4%, *n* = 3891) went on to purchase the originally intended codeine product. Around two-thirds of respondents were keen to receive further professional (*n* = 15) and clinical (*n* = 14) training to help better support customers in their use of OTC codeine.

Improved patient knowledge was reflected in the identification of risk minimization options; 78.3% of respondents (*n* = 18) supported a patient education and awareness campaign and 65.2% (*n* = 15) supported the cessation of promotional activity and advertising of OTC codeine. Only 47.8% (*n* = 11) supported a switch (upschedule) of codeine to a Prescription-Only Medicine.

## 4. Discussion

These results demonstrate that ease of utilizing the card was linked to the staff-perceived receptiveness of each customer. Despite the differing levels of customer engagement, this supports previous research [48] that some consumers will be seeking advice from pharmacy teams. With the majority of interactions using the card classed as easy and with receptive customers, this work indicates that pharmacy teams can influence an individual’s knowledge and comprehension about the safe and appropriate use of OTC codeine. Due to the variable skills and expertise of pharmacy teams previously identified [41], the requirement for training to enhance the interactions remains [7].

Although the level of ease seemed related to the receptiveness, it is not possible to determine from the data if one precedes the other, i.e., if an easy conversation leads to a receptive customer or whether it is the receptiveness of the customer that determines the ease. Furthermore, staff may have categorized the ease and the receptiveness differently based on their own ability (particularly with respect to the subjective nature of perceived ‘receptiveness’) and their perception may have been altered by, for example, familiarity with customers or language barriers. This may provide some explanation as to why some conversations deemed very easy still resulted in the customer being perceived as unreceptive or refusing the card. Further work to understand this could be considered with more robust operationalization of the idea of receptiveness.

Advertising of OTC codeine is considered to add weight to the perception of safety and efficacy, and could result in customers seeking to purchase the product [49] even where staff consider it may not be the most appropriate treatment. Given cessation of advertising was ranked high as a risk minimization option in the survey, it is apparent that pharmacy teams wish to persist in their role as the prime source of advice for consumers on use of OTC medicines and be able to influence their health choices.

Despite other research showing that previously used medication and medicines advice can support appropriate product selection [37], staff remained of the view that the most appropriate outcome was only achieved in around one third of interactions. Coupled with tracker data showing that the vast majority of customers continued with their intended purchase, this validates previous research that customers seeking OTC codeine can be better supported [7,50]. This type of interaction with pharmacy staff provides potential opportunities for early intervention where pain management and the use of opioids is concerned; reinforcing appropriate use and helping to reduce ‘gateway’ risk. It further supports the principles of joint decision-making and patient empowerment.

The most appropriate outcome is, however, subjective and based on judgement, particularly in the absence of a formalized clinical relationship between the pharmacy team and the consumer. Nonetheless, these results do show that the card appeared to influence the outcome for some customers and, combined with the survey responses noting how it would encourage customers to consciously consider their choices, this reveals that the card impacted customers’ reflective motivation. This reflective motivation could then have been mediated by the information on the card, with key risks presented in a novel way, affecting customers’ psychological capability.

Staff were keen to recommend the card to colleagues and see enhanced patient education and awareness in relation to OTC codeine although this assumes customers do not already have the requisite knowledge. The heterogeneous intentions of customers with regards to seeking medicines advice [51,52] does, however, reinforce the role of community pharmacy in fulfilling those needs [48], particularly as customers can opt to purchase OTC codeine online thereby avoiding a face to face interaction.

A small proportion of customers were immediately nudged into an alternative outcome and, despite not being observable during the trial period, it is possible that the impact of the nudge affected additional customers at a later stage, once they had an opportunity to absorb the card information [36,53]. The results also reveal two important behavioral changes for known customers, where the purchase of OTC codeine may have become habitualized. First, known customers were more likely to refuse the card and, second, they were more likely not to make a purchase for a pain relief medicine following intervention with the card. This demonstrates that the card disrupted the normal course of their behavior. One explanation is that the customer responded to the card, but equally they could have felt embarrassed or stigmatized with the interaction, or have chosen to purchase OTC codeine elsewhere. The effect observed could have been greater as identification of 8.7% of customers as known customers is considered to be an understatement of the reality based on previous research [15,17]. This may be due to staff being unwilling to categorize customers in this way, lack of awareness, or a reluctance to admit that previous counselling efforts have been ineffective as witnessed by others [41]. It is also possible that Covid-19, given its significant impact on reducing the overall number of customers attending pharmacies, may have distorted the known customer proportion and may have also caused wider variation in transactions for each pharmacy than may have been expected. This, along with the high pressures faced by pharmacies during this time, is also likely to have been the cause of a proportion of pharmacies not returning any data.

The benefit to unknown customers can also be seen by their perceived receptiveness to the card. For these customers, the card can deliver an early intervention into codeine use, and potentially amend future behavior when it comes to purchasing OTC codeine [26]. It also provides an opportunity to signpost and refer those needing additional help onto other more appropriate services.

The main limitations of this study design are that it is not possible to compare the data to a baseline as the potential outcome without use of the card is not known, nor is the number of customers purchasing OTC codeine during the data collection period known. In addition, without access to medical history, or a robust referral mechanism to an alternative healthcare provider or service, the ability of pharmacy teams to provide comprehensive care is limited. Although these results indicate that 10% of interactions were with known customers and this may raise concerns regarding their care, it is not clear whether this is an accurate estimate of this population. There are a number of caveats that need to be considered when contemplating this. For example, staff turnover, shift patterns and use of locum workers will all impact the consistency of staff a customer may deal with and Covid-19 may have impacted the types and mix of customers presenting in the pharmacy. In addition, it was not the objective of this study to quantify the percentage of known customers, but to understand their receptiveness to the support, therefore the method of tracking was not designed to do this robustly. This does highlight that the needs of these customers may be different to others and that community pharmacy should be able to identify these customers and direct them to specialist services. These limitations could be addressed in further research. Despite these limitations, given the exploratory nature of the project, the geographical spread and high number of interactions recorded, these results still provide key insights into understanding how an educational intervention could be used in community pharmacy to nudge consumer behavior.

## 5. Conclusions

The findings indicate that the card was considered by staff to be easy to use and a useful prompt to support discussions with customers on appropriate and safe use of codeine-containing medicines. It enhanced the information provided during normal medicines counselling, was perceived as being generally well-received by customers and, in particular, supported conversations with customers that were known to the pharmacy staff (resulting in a higher proportion of these customers having an outcome which differed to the originally intended purchase). These results demonstrate the value community pharmacy can bring by being able to change consumers’ behavior and that this can be delivered with a physical, visual, low-cost product to augment the verbal advice and counselling currently provided. If implemented sector-wide, the behavioral change effect could be even more marked. With staff training to boost confidence and moral courage to tackle misuse, the card can add to public awareness of the risks associated with codeine and support retention of codeine as an OTC medicine.

## Figures and Tables

**Table 1 pharmacy-08-00185-t001:** Ease/difficulty of conversation and perceived receptiveness of known and unknown customers (Note: each table adds up to 100% to show distribution across cells).

**Known Customers (*n* = 347)**				
**Receptiveness**	**Ease/Difficulty**			
	Very easy	Quite easy	Quite difficult	Very difficult
Refused the card	5.2%	1.1%	1.4%	3.2%
Not receptive	7.8%	3.2%	4.0%	1.2%
Quite receptive	19.0%	7.2%	0.0%	0.0%
Very receptive	45.0%	1.7%	0.0%	0.0%
**Unknown Customers (*n* = 3646)**				
**Receptiveness**	**Ease/Difficulty**			
	Very easy	Quite easy	Quite difficult	Very difficult
Refused the card	1.1%	2.5%	1.8%	1.1%
Not receptive	5.5%	5.8%	4.3%	0.5%
Quite receptive	17.8%	14.2%	0.9%	0.0%
Very receptive	42.4%	2.0%	0.1%	0.0%

**Table 2 pharmacy-08-00185-t002:** Outcomes of interactions using the card.

Outcomes	Unknown Customers (*n* = 3646)	Known Customers (*n* = 347)	Total (*n* = 3993)
Original codeine product purchased	97.7%	94.5%	97.4%
Smaller pack of codeine product purchased	0.6%	0.0%	0.5%
Non-codeine product purchased	0.6%	0.0%	0.6%
No product purchased	1.0%	5.2%	1.4%
No product purchased and customer signposted	0.1%	0.3%	0.1%

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
