# Peer review of "Over-The-Counter Codeine: Can Community Pharmacy Staff Nudge Customers into Its Safe and Appropriate Use?"

_pharmacy, 2020, doi:10.3390/pharmacy8040185_

Round 1

Reviewer 1 Report

  • Introduction
    • Overall, I think this is a good intro but I have a few questions/points I think you could clarify that could give your background more context for the reader.
      • On average, what is the frequency of OTC dispensing of codeine? Is it kept behind the pharmacy counter? What needs to be logged for each purchase? Is there a daily/monthly limit on purchasing?
      • Is there any data on OTC codeine misuse? I know you stated it is a gateway drug but how common is this?
      • Clarify further on the recommended dosing/duration/package sizes that are available OTC
    • I think it would be useful to go into a little detail about the use of codeine in children and any recommendations on limiting it especially in children <12 due to increased risk for respiratory depression
      • https://www.fda.gov/drugs/drug-safety-and-availability/fda-drug-safety-communication-fda-restricts-use-prescription-codeine-pain-and-cough-medicines-and
    • The first sentence in the paragraph beginning on line 64 should be reworded (lots of negatives and it is a run-on).
  • Materials and Methods
    • Clearly state your objectives
    • Line 92- you state that the stores are chosen based on volume of codeine sales. What is the average volume? How does this compare to other stores? I note this because in the US, although codeine is available OTC, it is hardly dispensed in this way.
    • Many questions about your methods that could help the reader to better understand how your study was set up:
      • Who had the card? How was it distributed?
      • Who did the tracking?
      • Who did the counseling? Is there any potential for confusion if a tech completes the counseling instead of a pharmacist?
  • Results
    • It seems there may be something off with your math on lines 117-118. You state that there are 8-480 responses overall and then 1-48 per pharmacy per day. If the study duration was 5 weeks, and the if there was truly 1 response per pharmacy per day, the minimum would be 35 (not 8)...so it seems the range for per day should include 0?
    • Tracker data
      • Why did you separate out known customers from unknown in the presentation of your data? This was not listed as part of the purpose. (May be an objective but we need those to be listed clearly).
      • One thing that I have a little issue with is your presentation of "known customers." In the manuscript, it makes it seem as these are people that the pharmacy normally/often cares for, but on the survey it states that they are known for purchasing codeine. These are two very different scenarios. Please clarify.
    • Staff survey
      • How many did you distribute, who replied (pharmacist vs. tech, etc.)?
  • Discussion
    • Consider removing the mention of COVID (line 214-216) as it does not seem to have impacted your N or your outcome.

Reviewer 2 Report

This is a welcome if modest attempt to explore a brief intervention for individuals visiting community pharmacies to purchase codeine-containing analgesics. The paper is generally well written and offers a helpful introduction although see comments below about whether all the claims and references are appropriate; I did wonder if the Cooper 2013 paper on 'Surveillance and Uncertainty' in HSCC might have been useful to cite for readers wanting to know more about UK pharmacy staff strategies and difficulties around opioids and other medicines of misuse. The one key aspect of the introduction to review is that the behaviour change aspect is very brief and although it is really welcome to see this used as the basis for an intervention, readers might need more orientation to what COM-B is, and to nudge approaches (and this does link to a methods point below also). The methods are quite well described if brief and arguably there are some issues around the sampling and operationalising of concepts and the study does rely on a somewhat skewed sample of pharmacy sites and self-report constructs. However, some of this is recognised in the paper and readers will have appropriate caution in several important areas. The findings were again quite brief but conveyed a good level of detail, and the discussion linked well to other studies.

Overall, this is worthy of publication despite being a somewhat modest pilot study (and on that point, it will be important to review how this study is described as it appears to be considered 'service evaluation' for the purposes of whether ethical approval was needed (assuming not if SE) BUT the paper variously refers to this as a pilot and research is used 8 times to refer to the study.

More specific comments (numbers refer to lines):

22 - UK needs written in full

30 - 'more likely not to purchase' - is an exact figure available to give readers?

36-40 - need to clarify that codeine is co-formulated OTC and in fact this leads to more of a clinical issue (ie hepatotoxicity and GI problems etc).

45 - reference 7 refers to codeine occasionally but I could not see it explicitly referring to codeine being a gateway drug; can this be checked?

53-56 - good to consider the information sources but I wondered if more could be said about this as the Cooper Respectable Addiction study did highlight this problem of where advice and help could be found. The internet is also an interesting area to consider, and there is a recent paper that again shows how people seem to bypass health professionals (shame/stigma as you note?) and for example use internet forums, which I wonder if readers would value in linking to (Lee and Cooper 2019, Codeine addiction and Internet Forum Use and Support JMIR Mental Health 6(4), e12354).

68 -  if reference 24 actually about Rx codeine and not OTC. Please check.

68 - 'some experts' but only one reference.

70-76 - would the Cooper 2013 Surveillance and Uncertainty paper in HSCC be useful to readers here?

79-80 - why capital letters initially?

77-84 - more on behaviour change and COM-B (and links to methods as it is not clear how the theory was actually used to develop the card and content)

85 - 'aim of this evaluation was to pilot' - seems a contradiction of what a service evaluation is. See above and clarify. I actually think it is a pilot intervention (and indeed it is noted that there was no pre-intervention baseline data which seems to suggest this was considered) but depending on what research ethics status this project had, we need to be careful.

89-90 - how were the 5 messages derived. The BNF entry contains a lot of helpful information so how was only some of it selected (eg expert panel devided etc perhaps?) BUT of note is that some the messages don't seem to be in the BNF monograph and this nees clarifying. I'm fine with them and they are plausible BUT for a credible design, there needs to be transparency about how they were selected, eg for other researchers using similar methods.

92 - why only high codeine volume stores? I understand that this would maximise the number of interactions in the brief 5/52 BUT can the biases introduced by this be considered further; also, were these high codeine sales by absolute sales or relative to the footfall etc (ie are they supplying more codeine analgesics because they are just high volume, high footfall stores, OR are they supplying more codeine relative to their other sales); both are a possible bias and just need more clarification.

94-95 - this is interesting re the existing Boots framework, and readers (and other researchers) would benefit from knowing more about this, as one argument is that for 'unknown' customers, it is not just the card but perhaps the Boots framework that could be influencing their behavioural choices.

99 - 'codeine product' - clarify if staff were told only analgesics (ie not codeine linctus), and also presumably not dihydrocodeine in Paramol also).

99 - tracker - was this paper or electronic, and how was data entered

103 - 'known customer' as someone who frequently purchases codeine is obviously very subjective and do we know if the staff might have been locums or not as familiar with the store and hence could not judge. Does this also raise a slight concern as to how they are being managed apart from this intervention; this study does seem to (somewhat unintentionally) show that around 10% of customers were regulars and is this something to consider? ie should staff be supplying them and creating this regular status? It's a fascinating and vexed issue and it is tapped into in the discussion and I liked the known/unknown aspect (see the Cooper Surveillance and Uncertainty paper again for this).

104-105 - this part was very brief and the associations for example were not stated.

106 - was any piloting of the survey (or indeed card) ever done?

116-119 large variation and especially since the stores were chosen as they were high codeine sales. Any reasons to offer readers.

116-119 12 did not participate at all. Any insights into why non-engagement occurred might be of use (which I know all too well in such research!)

119 - only 23 survey results so considerable caution needed in this data but it is treated quite modestly. Do we know if this was a pharmacist or MCA who completed? Shame this could not have been done as phone interviews instead given the small sample and more qualitative insights.

167-173 - agree and the subjective nature of 'receptiveness' is important to highlight, and a furether study might want to operationalise this more robustly.

209 - agree the issue of stigma/embarrassment/shame is a really interesting one and the Cooper Respectable Addiction report and also the Lee and Cooper Codeine Addiction and Internet Forum paper in JMIR Mental Health explores this too.

I hope these comments are of help and this is a welcome, if small scale and modest project which I think could form the basis of a much more robust intervention study; a final point is that in a way this study (somewhat inadvertently I suspect) highlighted issues around 'known' regular customers and raises some questions as to why this and their associated status is perpetuated BUT importantly how as a group their behaviour is different to 'unknowns' and this is a fascinating emerging aspect.
